# Small Non-Coding RNAs Derived from Eukaryotic Ribosomal RNA

**DOI:** 10.3390/ncrna5010016

**Published:** 2019-02-04

**Authors:** Marine Lambert, Abderrahim Benmoussa, Patrick Provost

**Affiliations:** CHU de Québec Research Center/CHUL Pavilion, 2705 Blvd Laurier, Quebec City, QC, G1V 4G2 and Department of Microbiology, Infectious Diseases and Immunology, Faculty of Medicine, Université Laval, Quebec City, QC G1V 0A6, Canada; Marine.Lambert@crchudequebec.ulaval.ca (M.L.); Abderrahim.Benmoussa@crchudequebec.ulaval.ca (A.B.)

**Keywords:** biogenesis, microRNAs, ribosomal RNA-derived fragment (rRF), ribosomes, small ribosomal RNA (srRNA), ribosomal DNA (rDNA), small RNAs

## Abstract

The advent of RNA-sequencing (RNA-Seq) technologies has markedly improved our knowledge and expanded the compendium of small non-coding RNAs, most of which derive from the processing of longer RNA precursors. In this review article, we will present a nonexhaustive list of referenced small non-coding RNAs (ncRNAs) derived from eukaryotic ribosomal RNA (rRNA), called rRNA fragments (rRFs). We will focus on the rRFs that are experimentally verified, and discuss their origin, length, structure, biogenesis, association with known regulatory proteins, and potential role(s) as regulator of gene expression. This relatively new class of ncRNAs remained poorly investigated and underappreciated until recently, due mainly to the a priori exclusion of rRNA sequences—because of their overabundance—from RNA-Seq datasets. The situation surrounding rRFs resembles that of microRNAs (miRNAs), which used to be readily discarded from further analyses, for more than five decades, because no one could believe that RNA of such a short length could bear biological significance. As if we had not yet learned our lesson not to restrain our investigative, scientific mind from challenging widely accepted beliefs or dogmas, and from looking for the hidden treasures in the most unexpected places.

## 1. Introduction

### 1.1. Ribosomes

The ribosomes are ribonucleoprotein (RNP) complex assemblies required for the translation of all proteins [1,2]. During eukaryotic evolution, ribosomes have considerably increased in size, forming a surface-exposed ribosomal RNA (rRNA) shell of unknown function [3]. This surface may be an interface for interacting proteins, as suggested by the identification of hundreds of ribosome-associated proteins (RAPs) from categories including metabolism and cell cycle, as well as RNA- and protein-modifying enzymes that functionally diversify mammalian ribosomes [4]. rRNA sequences may also be modified, as the presence of ufmylation suggests [4], or be cleaved to form new functional small non-coding RNAs (ncRNAs) species. Therefore, the interplay between RAPs, biochemical changes, and generation of new small ncRNAs may provide an additional layer of regulation and govern one of life’s most ancient molecular machines involved in protein expression [5,6].

### 1.2. Ribosomal DNA Genes

Mammalian ribosomal DNA (rDNA) genes code for rRNAs that are generally comprised of several hundreds of transcription units organized in tandem repeats and clustered on a number of chromosomal loci [7,8,9]. For example, in humans, there are approximately 300–400 rDNA repeats in five clusters on chromosomes 13, 14, 15, 21, and 22 [8], potentially explaining the rRNA sequence variability. This variability has notably been reported by Wang et al. [10], who calculated the coefficient of variation (CV) of the average depth among samples in the 18S and 28S rRNA, which they reported as a diagram of variability as a function of the sequence region [10]. Although the rRNA sequence of the 18S, 5.8S, and the 28S may contain variations, consensus motives can be found [11,12]. The rRNA secondary structure is also extremely well conserved in eukaryotes, thanks to a strong selection pressure [13,14]. Being the main component of ribosomes and acting at the core of their function [15], rRNA expression is finely controlled, from the regulation of transcription to their biogenesis [7,16,17,18]. Other processes involved in the rRNA maturation can also be regulated, such as biochemical modifications of their RNA sequence [19,20,21] or incorporation into the ribosomal complex [16,22]. 

### 1.3. Ribosomal RNAs

Each ribosome is composed of rRNA—constituting its functional core—and ribosomal proteins. In eukaryotes, ribosomes consist of four rRNAs and approximately 80 ribosomal proteins arranged into two subunits: 60S and 40S [23]. During the ribosome biogenesis process, eukaryotic rRNAs are usually synthesized by the RNA polymerase I (RNA Pol-I) in the nucleolus, giving rise to a single rRNA precursor: the 45S rRNA [24]. This long primary transcript contains several different rRNAs separated by spacer regions, known as internal transcribed spacers (ITS; ITS1 and ITS2). Indeed, three of the four mature rRNAs (18S, 5.8S, and 28S rRNAs) originate from this common 45S precursor, while the 5S rRNA is synthesized by RNA polymerase III (RNA pol-III) [25,26]. 

#### Ribosomal RNA Sequence Variants

Nevertheless, it has been shown that distinct ribosomal genes (rDNA) are expressed at different stages of development, leading to the incorporation of alternative forms of rRNA with heterogeneous sequences into ribosomes [27,28]. It is admitted that embryogenesis is a tightly regulated process, in which the control of expression of specific proteins, at key embryonic stages, is important [29]. Thus, the heterogeneity found in ribosomes may allow this regulation by favoring translation of specific sets of messenger RNAs (mRNAs) into proteins. Both the type of proteins present in the RNP complex [4,23,30,31,32] and the rRNAs may contribute to this regulation, as in some alternative pathways during the eukaryotic rRNA maturations [22,24,33]. In addition, splicing of some rRNA transcripts may be decoupled and lead to the production of new rRNA intermediates (e.g., 43S and 26S) [34]. Furthermore, the first rRNA variants described are at the 5′ end of the 5.8S rRNA [35]. Thus, two forms of 5.8S rRNA exist—the 5.8S short (5.8SS) and the 5.8S long (5.8SL)—which differ in size by an extension in the 5′ end of 7 or 8 nucleotides (nt) [16,32,36]. Accordingly, similarly to ribosomal proteins, some of these rRNA variants may play a role in ribosome heterogeneity [21,32,37].

### 1.4. Small Regulatory Non-Coding RNAs

Since the discovery of the first small silencing RNA in 1993 [38], a noteworthy number of small RNA classes has been discovered, including microRNAs (miRNAs) [39], small interfering RNAs (siRNAs) [40,41], and Piwi-associated small RNAs (piRNAs) [42,43], all of which exerting important roles in various biological processes [44]. More recently, additional new classes of small non-coding RNAs have been discovered in the wake of the next-generation sequencing (NGS) revolution [45], which markedly expanded our knowledge of small RNAs. For instance, a class of small RNAs originating from small nucleolar RNAs (snoRNAs) has been identified to function like miRNAs [46]. Such studies have fueled interest in small RNAs that could derive from other non-coding RNAs, such as transfer RNA (tRNA) [47,48,49,50], snoRNA [46,51,52], and rRNA [53]. The importance of ncRNAs in cellular regulatory mechanisms, especially during ribosome biogenesis, and their contribution to ribosome heterogeneity, both compositional and functional [21], raised a particular interest in this field of study. 

These studies range from the small RNAs like the snoRNA [19] (HBII-95, HBII-234, etc.), which contribute to the rRNA maturation, notably by inducing biochemical changes on the rRNA sequence, to the large RNAs like the nucleolar-specific lncRNA (LoNA), which can suppress rRNA transcription and reduce rRNA methylation [54]. These two examples illustrate how, by changing rRNA methylation level, ncRNAs can modulate ribosome biogenesis and contribute to the ribosome heterogeneity by acting in specific environments (localization and times) [6,55].

### 1.5. Small Ribosomal RNA-Derived Fragments

In mammalian cells, ribosomal ribonucleic acid (rRNA) (~80%), mRNAs (~5%), and transfer ribonucleic acid (tRNA) (~15%) are the most abundant RNA molecules. Despite their relatively high enrichment and potential function, the small ribosomal RNA-derived fragments (rRFs) are usually removed as a by-product of RNA degradation from RNA-sequencing (RNA-seq) or small RNA-seq analyses [56,57]. Similarly, the full-length rDNA sequences are not included in human and mouse genome assembly, which represents an important gap in genome information [9]. 

However, even when rRNA sequences are included in the analyses, some rRFs may be missed because standard approaches use a reference transcriptome that does not include all the possible variations from rDNA sequences. To detect rRFs that differ from the rDNA sequences of reference, the challenge will be to annotate all repeat variability and existing SNPs (single-nucleotide polymorphisms) [10]. Notably, the 5S and 45S rDNA arrays display remarkable variability in terms of copy number (CN), ranging from tens to hundreds of copies among eukaryotes [58,59,60,61], and a 10-fold variation among individuals in human populations [58,59,60]. This information may be accessed by calculating the coefficient of variation (CV) of the average depth among samples, and allow comparative analyses between loci.

Nevertheless, over the past few years, scientists have begun investigating the existence, role, and function of specific small rRFs, which will be the topic of this review article. Whereas rRNA plays a role in ribosome heterogeneity, rRFs may be involved in the control of translation, albeit not excluding other important biological functions. Here, we will discuss about the discovery, biogenesis, protein-binding capacity, and function of small ncRNAs derived from rRNA. 

## 2. Small RNA Derived from Ribosomal RNA

### 2.1. Fragments of the 28S Ribosomal RNA

#### 2.1.1. Discovery: Cleavage, Localization, and Expression Pattern

The 28S rRNA is the longest rRNA and forms the large subunit (LSU) of eukaryotic cytoplasmic ribosomes [62]. The mature 28S rRNA is generated from the 45S rRNA upon cleavage into 32S pre-rRNA, which finally matures into the 28S rRNA after endonucleolytic and exonucleolytic processing [34]. Mature 28S rRNA may also be produced through the endonucleolytic processing of the 41S rRNA to a 36S intermediate [63]. Since they are mediated by endo- and exonucleases, which are often implicated in the generation of small ncRNAs, these parallel processing pathways might lead to the generation of functional and biologically relevant small rRNA-derived ncRNAs.

Wei et al. [53] have shown that up to 64 to 70% of rRFs were distributed to the human rDNA region encoding the 28S rRNA, as compared to 16.1–22.4% and 4.5–7.0% for the 18S and 5.8S, respectively. Therefore, the majority of rRFs mapped to the 28S rRNA, which is consistent with its larger size. In this case, one would expect the rRF sequences to be randomly distributed along the 28S rRNA. So, when Chen and collaborators found that rRFs were significantly enriched at the 5′ and 3′ ends of the 28S rRNA gene, in the *Amblyomma testudinarium* model and in human cell lines [64], the hypothesis that rRFs were generated by a specific endonucleolytic cleavage process, rather than a random exonucleolytic digestion [64], gained more credibility. The authors annotated a total of 26 rRFs, ranging from 15 to 40 nt in length, from both the 5′ and 3′ extremity of the 28S rRNA from *Amblyomma testudinarium* [64]. They are called rRF3, the rRFs originating from the 3′ end, and rRF5, the small non-coding RNAs found in the 5′ end of the rRNA. Surprisingly, the rRF3 series were more highly expressed than the rRF5 series, with a maximum rRF count of 5407 for rRF5s and of 1,433,580 for the highly expressed rRF3s [64]. Moreover, they demonstrated the biological significance of one specific rRF3 in human cells [64].

The 28S rRNA may also be the subject of atypical processing events, and give rise to known classes of small ncRNAs. In 2013, a study revealed that a number of noncanonical miRNAs mapped to ribosomal RNA molecules, with 1% of annotated miRNAs mapping to mature rRNA sequences [36]. Whereas mmu-miR-2182 originates from the 45S rRNA precursor, mmu-miR-5102, mmu-miR-5105, mmu-miR-5109, and mmu-miR-5115 are produced from 28S rRNA [39]. In mice, a total of 10 miRNAs are rRFs and 62 rRFs perfectly match piRNA sequences, including piR-16, piR-38, piR-170, and piR-171 (Figure 1 and Figure 2) [65]. Therefore, these findings—including the overlap of rRFs with miRNAs and piRNAs—support the idea that rRFs could be a functional small RNA.

The first small ncRNA species known to derive from the 28S rRNA was discovered in the filamentous fungus *Neurospora crassa* in 2009 [66], when DNA damage was found to induce their expression, together with the Argonaute protein QDE-2. Known as qiRNAs (QDE-2-interacting small RNAs) because of their interaction with QDE-2, these small RNAs are ~20–21 nucleotides long (several nucleotides shorter than Neurospora siRNAs), with a strong preference for uridine at the 5′ end, and originate mostly from the ribosomal DNA locus [66]. QDE-1 and QDE-3 proteins, together with OsRecQ1 and OsRDR1, are required and play critical roles in qiRNA biogenesis [66,67]. qiRNAs have been shown to mediate gene silencing in the DNA damage response (DDR) pathway, and are induced by DNA-damaging agents ethyl methanesulfonate (EMS) and UV-C (Figure 1). qiRNA expression has been reported to be affected in diabetes [53], where unique and redundant reads of rRFs peaked at different sizes for normal samples compared to the diabetic ones [53].

In plants, other rRFs, called phased small interfering RNAs (phasiRNAs), have been discovered (Figure 2) [40]. They are normally regulated by miRNAs [40]. The study reporting their existence revealed that some LSU-rRNAs (28 and 5.8S) could also generate phasiRNAs, suggesting that some rRNAs may be processed through the phasiRNA biogenesis pathway [40].

Finally, the longest rRF originating from the 28S rRNA, as reported among other rRFs in an RNA-seq study of Zebrafish development, measures 80 nt [27,28]. This 80 nt rRF, known as rRF3, maps to the 3′ end of the 28S rRNA sequence. Notably, rRF3 is relatively more abundant in the egg and adult tissue, compared to other embryonic stages [27,28] and differ in five nucleotides. 

#### 2.1.2. Sequence, Length and Structure

Previously described qiRNAs (*Neurospora crassa*) are approximately 20–21 nt in length and form a hook structure [66]. For phasiRNAs (plant), 50% of the 21-nt PHAS loci are in rRNA or repeats, and five are annotated as LSU-rRNA (Figure 1) [40]. In Wei et al. (2013), the RNA-seq data analysis from human samples revealed that the most abundant rRF was 21-nt long [53]. Thus, in most of the models and studies, the length of most rRFs is ~20–21 nt, a size comparable to miRNAs and other classes of small RNAs (e.g., piRNA, qiRNA, siRNA, and tRNA-derived RNA fragments (tRFs). 

The 28S rRNA sequence giving rise to rRF3 is involved in a stem-loop structure; a small rRF3 ncRNA can thus reverse-complementarily bind to the 3′ end of another complete 28S rRNA molecule [27]. This mode of recognition may regulate the stability and expression of the 28S rRNA, or favour the formation of RNA duplexes that are more susceptible to cleavage by endonucleases (e.g., Dicer) along a process that produces new small rRNA-derived ncRNAs. On the other hand, this rRF3 alone may be protected from degradation thanks to its stem-loop structure [27]—like transcripts with an intrinsic terminator are protected from 3′–5′ exonuclease digestion due to the stem-loop structure. These energetically stable stem-loop structures, which were not followed by uridine rich sequences, may also protect Rho-dependent transcripts from 3′–5′ exonucleases, such as PNPase and RNase II [68], and explain why this rRF is not degraded.

Finally, the two rRFs (maternal and somatic) reported by Locati and al. [28] (Zebrafish) exhibit major differences in terms of primary sequence and secondary structures, suggesting that they may be processed differently, associate to different ribosomal proteins and base pair with different mRNAs. As explained above, this could be part of the mechanisms underlying ribosomal heterogeneity and differential translation regulation.

#### 2.1.3. Function and Protein Binding

Short hairpin RNA (shRNA)-induced depletion of the 28S rRF3, in the H1299 cell line, significantly increased cell apoptosis and inhibited cell proliferation [64]. Moreover, rRF3 depletion resulted in a significant decrease of H1299 cells in the G2 phase of the cell cycle. Although the mechanisms involved in rRF5 and rRF3 biogenesis remain unclear, these results support the functionality of rRFs [64].

QDE-2-interacting small RNAs have been shown to be required for the DDR and repair pathway in rice [66,67]. In another study, qiRNAs from 28S rRNA were very closely related to piRNAs, and potentially work as small guide RNAs (Figure 1) [69]. The differential expression of the qiRNAs in diabetes samples suggests their possible involvement in the pathophysiology of the disease. Wei et al. [53] showed that the overexpression of these particular rRFs could impact expression of the key gluconeogenic enzyme genes PEPCK and G6Pase, by modulation of their promoter activity and also that of peroxisome proliferator-activated receptor gamma (PPAR-γ), which regulates lipid and glucose metabolism. Furthermore, the authors described a negative effect of these rRFs on intracellular ATP level, which is also downregulated in patients with type 2 diabetes. These results suggest that rRFs may participate to biological processes related to metabolic diseases [53]. An involvement of these rRFs in multiples pathways as p53 signaling pathway or other pathways involved in p53 upregulated modulator of apoptosis (PUMA) transcriptional activation have been shown [53,70]. Moreover, they detected an effect of rRFs on extracellular signal-regulated kinases (ERK) pathway including the phosphorylation of ERK1/2, p90RSK, Elk-1, and p70S6K [53]. ERK pathway plays an important role in the transmission of cellular proliferation and developmental signals [71], then rRFs seems to modulate in a broad range of biological processes and signaling pathways.

Studies involving Argonaute (Ago) protein immunoprecipitation, followed by high-throughput sequencing, on various species, including Arabidopsis and Drosophila models and human cell lines, revealed that rRFs co-immunoprecipitated with Ago1 and Ago2 [72,73,74,75]. The size distribution of the rRFs bound to Ago proteins were mainly around 20–22 nt, suggesting that rRFs may be part of, and mediate their function via, Ago complexes [53], just like miRNAs. Notably the rRF expression profile and distribution patterns seemed to be tissue specific [53], suggesting that Ago•rRF complexes may be cell- or tissue-specific. 

Also similar to miRNAs, phasiRNAs encoded by PHAS play important regulatory roles by targeting protein coding transcripts in plant species [40]. Generally, phasiRNA could be associated with Ago proteins, to repress the translation or contribute to the mRNA target degradation. Like miRNAs, phasiRNAs serve as guide to recognize the target RNA (Figure 1). In this way, another 28S rRFs co-immunoprecipitated with tRNase ZL in human kidney 293 cells and could work as small guide RNAs (sgRNAs) for tRNase ZL in vivo as well as in vitro [69]. The existence of small RNAs derived from 28S rRNA with functional properties has been demonstrated in several studies, as discussed above. The small rRF ncRNAs have been raising significant interest among the scientific community, mainly because of the potentially high abundance of these small RNAs (comparable to that of their rRNA precursors) as well as their possible involvement in gene regulatory mechanisms. New studies have demonstrated the presence of a diverse array of RAPs on ribosomes [6,76] that may be capable of generating other rRFs and, as the example of phasiRNA [40], using rRFs into a RNP complex of regulation. 

### 2.2. Fragments of the 18S Ribosomal RNA

The biogenesis pathway leading to the formation of the 18S rRNA is different from that of the 5.8S and 28S rRNAs [77,78]; rRFs derived from the 18S rRNA may thus be generated through different mechanisms and have different regulation modalities [64]. 

#### 2.2.1. Discovery: Cleavage, Localization, and Expression Pattern

The most abundant maternal-type rRF detected during Zebrafish developpement [27] comes from the 5′ end of the 18S rRNA and measures 21 nt. The most abundant somatic-type rRFs, however, originate from the 5.8S rRNA, with some rRFs originating from the 18S rRNA. The most abundant rRF derives from the 5′ end of the 18S rRNA and is 130 nt long; it may either exert a function per se or be the precursor of the 21-nt rRF detected in the maternal-type, along a process resembling that of primary miRNAs giving rise to mature miRNAs [27]. 

Interestingly, a group of clustered noncanonical miRNAs derive from pre-rRNA (Figure 1 and Figure 2), and three of these miRNAs were mapped to the 18S subunit: pso-miR-2914, pso-miR-2916, and pso-miR-2910 [83].

#### 2.2.2. Structure, Localization, and Expression

The 130-nt rRF ncRNA derived from the 18S rRNA, as described above, has a secondary structure with a stem and a complex hinge with three smaller hairpins [27]. In fact, this rRF can form a stem-loop structure potentially similar to other functional ncRNAs, such as tRNAs [91] and snoRNAs [92], and represent a potential noncanonical miRNA precursor that may be further processed and loaded into an Ago protein [27].

#### 2.2.3. Function and Protein Binding

It has been reported [27] that both the guide and passenger strands of rRFs can associate with Ago proteins, suggesting that the 21-nt rRF RNAs may function like miRNAs and regulate gene expression [93]; as many as 532 putative target transcripts of rRFs have been identified [27].

The existence of rRNAs and of rRFs suggest dual molecular functions [39,65,83,94]. As reported for tRNAs [50], rRNAs may either function as mature rRNAs inside ribosomes or be processed into smaller fragments and act in a miRNA-like fashion. Indeed, such rRNA transcript units were shown to harbor as many as five different miRNAs, which, upon their release, are able to directly repress the expression of hundreds of genes at the post-transcriptional level.

Finally, these clustered miRNAs were differentially expressed in different tissues, suggesting that rRNA processing into rRFs may be placed under specific spatiotemporal control [83]. Expression of rRNA affects cell growth and proliferation, but the mechanisms that modulate rRNA levels are poorly understood. However, a new subset of 22-nt RNAs, called antisense ribosomal siRNAs (risiRNAs), show sequence complementarity to 18S and 26S rRNAs [95]. They were shown to act through the nuclear RNA interference (RNAi) pathway to downregulate pre-rRNA level [95]. Stress conditions, including low temperature and UV irradiation, induced the accumulation of risiRNAs which, in turn, may regulate pre-rRNA expression and represent a mechanism to maintain rRNA homeostasis [95] or induce expression of rRFs.

### 2.3. Fragments of the 5.8S Ribosomal RNA

The existence of two forms of 5.8S rRNA, with 7 or 8 different nt at their 5′ end, is widely described in eukaryotes [16,35,36,96]. Although the ratio between the two forms varies from one organism to another, the shorter form of 5.8S rRNA (5.8Ss) is predominant over the longer form (5.8SL), as it accounts for 80% of the total [34,77,96]. The short and long forms of 5.8S rRNA derive from different biosynthetic pathways, revealing the heterogeneity in the cleavage and processing of this RNA [16,62,97], which may lead to the release of small non-coding RNA fragments that have yet-to-discover biological roles. In this section, we will describe the small non-coding RNAs resulting from the 5.8S rRNA, their origin, sequence, and cleavage, but also the proteins they are associated with, their expression pattern, and their function. 

#### 2.3.1. Discovery: Cleavage, Localization, and Expression Pattern

RNA-seq data, obtained from developing Zebrafish [27], unveiled the existence of two distinct fragments of the 5.8S rRNA, which correspond to the 5′ and 3′ halves of the 5.8S rRNA. The rRF originating from the 5.8S rRNA 5′ end measures 75–76 nt or 74 nt, according to whether they come from maternal or somatic cells, respectively [28]. The length of the 5.8S rRNA 3′ end rRF is 74 nt in maternal cells and 81 nt in somatic cells [27,28]. These fragments are relatively long for non-coding RNAs; although they are longer than miRNAs [83], piwiRNAs [43], or tRNA fragments (tRFs) [47,50,98], they have a length similar to other small ncRNAs, such as snoRNAs [99]. Although these rRFs are longer than tRFs, which measure between 16 and 50 nt [100], they share similarities in their mode of cleavage. Indeed, tRFs have also been found to originate from cleavage of either the mature tRNA or the tRNA precursor molecule. In the latter case, RNase Z cleaves the 3′ part of the tRNA precursor as part of the maturation process, with the resulting fragment also being considered a tRF [47,101]. tRFs that are derived from mature tRNAs emerge after cleavage at either the D-loop (giving rise to 5′-tRFs) or the T-loop (giving rise to 3′-tRFs, with the CCA addition present). The same process may occur for rRFs, which may result either from mature rRNA or rRNA precursors [101]. 

Because the rRFs mentioned above are not the only ones originating from the 5.8S RNA in eukaryotes; some forms are shorter and more abundant [64,102,103], and are possibly generated by a process similar to the one described for miRNAs and involving one or more Dicer-like endoribonucleases. For instance, the highly abundant rRFs discovered in *Piper nigrum*^77^, originating from the 5′ end of the 5.8SL rRNA and representing the largest subset of rRFs of 23 nt [102]. 

The majority of the 20-nt long fragments deriving from the mature sequences of tRNAs, rRNAs, snoRNAs, and small nuclear RNAs (snRNAs), are produced, in a specific cleavage pattern, from the 5′ or 3′ end [103,104]. The 5′ or 3′ end origin seems to be different according to the tissues, development stages [27], or environment [104]. Li et al. [104] showed that the rRFs derived from 5.8S, 18S, and 28S rRNAs are generated upon cleavage of either the 5′ or 3′ end, with a preference for a 3′ end origin in human cells. Interestingly, most of the prominent clustered rRFs are coming from the 5.8S, rather than the 18S or 28S, rRNA [27,64,102], which is surprising given that the 5.8S rRNA is the shortest of the three. In plants, the 5′ end rRF cluster (rRF5) from the 5.8S rRNA is the most abundant, whereas the proportion of rRFs from the internal and 3′ end (rRF3) of 5.8S rRNA is much lower [102]. Similarly, in eggs and in adult tissues of Zebrafish, the 5.8S rRF5 is 3 and 4 times more abundant than the rRF3, respectively [27]. It differs from the rRFs of the 18S and 28S rRNAs, which are mainly produced by cleavage at the 3′ extremity [27]. In human cells and in ticks, rRF5 and rRF3, from both the 5.8S and 28S rRNAs, derive from either extremity, but more from the 3′ end [64]; the most abundant rRFs are 33 nt and 29 nt long, and belong to the rRF3 series [64]. The higher abundance of rRFs derived from the 5.8S rRNA, compared to the 18S and 28S rRNA-derived fragments, suggests that these rRFs may be protected from degradation and stabilized through their association with proteins. The relative abundance and cross-species conservation of the rRFs generated [64], as well as the bioactivity of RNA sequences of similar size, prompt the need for further investigations into the molecular and biological role and function of rRFs.

Interestingly, three regions of 5.8S rRNA can form helices by base pairing with the 28S rRNA as we can see in the secondary structure diagrams found in Andreas M. et al., supplementary data [105]. The longest interaction occurs between the 3′ end of 5.8S and the 5′ end of 28S rRNA [106], whereas the last two helices form between the 5′ end of 5.8S and the 3′ extremity of 28S rRNA. A possibility could be that these helices may be recognized and processed by endoribonucleases to generate rRFs from both 5.8S and 28S rRNAs. Similarly to tRFs, termini-specific processing and asymmetric stabilization could be observed for rRNAs, snoRNAs, and snRNAs [104,107]. The levels of rRF3 and rRF5 produced differ according to the species, environment, aging, or developmental stage.

A 5′ consensus 5.8S rRF sequence of 22 nt was found in all seed plants (Spermatophyte plants) [102]. The major cleavage was observed at the cytosine (C) in all species belonging to the Poaceae family. The rRF5 variants harboring a C as the terminal nucleotide ranged from 21 to 25 nt in length, and were the major form of rRF5 produced during pathogenic infection. Together, these results suggest that the 5.8S rRNA precursor may be cleaved at the C, to generate the small ncRNA rRF5, during the formation of 5.8S rRNA in pathogen-stressed plants. The rRF sequences produced by the 5.8S rRNA are also well defined and conserved, suggesting that rRF generation is a finely tuned process, and that rRFs may fulfill conserved function(s) as yet to be determined. Interestingly, tRNA-halves and tRFs have already been described in a stress-dependent phenomenon [107,108,109,110,111,112], involving translational regulation [113,114,115,116]. We can thus imagine that a similar phenomenon may occur for the 5.8S rRNA and lead to the production of rRFs under stress conditions [102,103].

A new small ncRNA, mapping to the 5′ region of rRFs and called cosRNAs (clustered organellar short RNA fragments), were recently identified (Plasmodium) [117]. These rRFs are between 18 and 40 nt in length, and could be generated by endonucleolytic cleavage along rRNA maturation. Importantly, cosRNAs do not exhibit a random distribution, as expected for a stochastic RNA degradation process, but were found specifically enriched at selected 5′ ends of rRNA. 

A large group of piRNAs and endogenous small interfering RNAs (endo-siRNAs), produced upon rRNA processing, have been identified as “unconventional” small ncRNAs with regulatory functions in mouse [118]. These rRFs are particularly abundant in spermatozoa. Notably, the most abundant rRF comes from the 5′ end of the 5.8S rRNA, measures 23 nt in length and is present at >4 million reads in the spermatozoa samples [118]; such an abundant rRF likely plays a role in mouse gametes.

The 5.8S rRNA participates to ribosome translocation and thus exerts an essential role in protein synthesis [15,119]. The formation of 5.8S rRNA may give rise to rRFs that may regulate 5.8S rRNA function in mRNA translation, by (i) interfering with the liaison between the 5.8S and the 28S rRNAs, (ii) impairing the function of ribosomal proteins, and/or (iii) exerting a different function as part of another RNP complex. The abundance of 5.8S rRNA in cells and the different pathways involved in its processing are expected to yield relatively high levels of rRFs, which may mediate important regulatory functions, and possibly contribute to ribosome heterogeneity by interacting with the translation machinery elements [118,120,121,122]. 

#### 2.3.2. Structure, Localization, and Expression

In silico analysis of *Arabidopsis thaliana* 5.8S rRNA predicts a secondary structure composed of hairpins and of a noncanonical miRNA-like short hairpin precursor, to which the second most abundant class of 5.8S rRFs could be mapped [102]. Locati et al. (Zebrafish) [28] discovered that the cleavage site lies in a loop at the exact location where the maternal-type 5.8S rRNA sequence has an AC insertion, as compared to the somatic one [27]. This process is similar to the cleavage of the tRNA anticodon loop, by an endoribonuclease, yielding tRNA 5′ and 3′ halves [123,124,125]. Lately, the 5′ and 3′ halves, resulting from the 5.8S rRNA cut, were found to display rather strong secondary structures, showing long stable stems [27]. Once the 5.8S rRNA is cut, the 5′ half (rRF5) has only two 28S rRNA binding regions, and the 3′ half (rRF3) one. Regarding these two or one potential binding region between the two rRFs and the 28S rRNA, we could imagine a competition effect of these rRFs on the 5.8S and 28S rRNA hybridation. For example, this competition may slow down the LSU speed association.

Concerning expression patterns, the phenomenon of 5′–3′-specific processing is observed across all major classes of ncRNAs, except mRNAs [104]. Moreover, these rRNA-derived sequences were more abundant than snoRNAs and snRNAs, but less abundant than tRFs in human and mouse cells [104]. Several observations indicate that cleavage of tRNAs and rRNAs is induced by various stresses [103,107,113,114,115]. Wang et al. [103] found that 8822 small ribosomal RNAs (srRNAs) were responsive to heat stress, and that production of sRNAs from tRNAs, 5.8S rRNAs and 28S rRNAs was more specific than that from the 5S rRNAs and 18S rRNAs (wheat) [103]. Although maternal-type 5.8S rRNA is degraded during the late stages of embryogenesis, the level of 5.8S rRFs is relatively unaffected, suggesting that these rRFs are stabilized.

#### 2.3.3. Function and Protein Binding

RNA-seq analysis of Ago co-immunoprecipitation experiments [126,127,128] in *Arabidopsis thaliana* and *Oryza sativa* revealed an association between 5.8S rRF5 variants and Ago complexes [102], such as Ago1 to Ago9 [102]. Specific Ago association may confer specialized function to 5.8S rRF5s, support their functionality, and suggests that they may have a gene regulatory role similar to miRNAs. However, unlike miRNAs, rRF maturation neither depends on Dicer or Drosha-mediated processing, nor does it rely on DGCR8 activity [104]. Interestingly, studies have found an association between rRFs and some proteins, such as PIWI proteins for qiRNA or piRNA, and their potential involvement in post-transcriptional regulation of mRNA transcripts (Figure 1) [44]. 

Nonetheless, the Ago-rRF interaction could have another explanation: would it be possible for rRFs to be the targets of a miRNA? For instance, a study identified chromatin-binding sites for Ago2 throughout the 45S region of the human rRNA gene [129]. This would mean that the human Ago2 would be tethered to ribosomal RNA through microRNA. Moreover, 479 experimentally verified regions were found within the 45S rRNA transcript and have the ability to form a perfect duplex with seed sequence of miRNAs expressed in HEK293T cells [129]. 

On one hand, this interaction could be dependent on Dicer, as the cross-linking of Ago2 to the rRNA was reduced to background levels in the absence of Dicer [129]. Moreover, Dicer was found associated to both active and inactive copies of the rRNA gene [129]. In the other hand, Ago2 is bound to a large number of rRNA-derived small RNAs that are synthetized in a Dicer-independent manner [128]. Interestingly, there was no change in Ago2-bound small RNAs derived from the rRNA region in the absence of Dicer [128]. This would mean that Ago2 interaction with rRNAs might depend on Dicer for the full-length rRNAs but could exist without Dicer for the already existing rRFs.

Together, these results suggest that rRFs may exert important functions in fundamental mechanisms through their association with specific proteins, as each rRF may exert a specialized function depending on its incorporation into specific RNP complexes. 

### 2.4. Fragments of the ITS1 and ITS2 RNAs

Deep sequencing analysis of small RNAs that emanate from the highly repetitive rDNA arrays of Drosophila revealed the existence of small RNAs deriving from internal transcribed spacer (ITS) of rRNA [65]. The authors also identified a novel, deeply conserved and widely expressed noncanonical miRNA mapping to the ITS1 region of rDNA [65], which was not identified previously due to bioinformatics filters removing such repetitive sequences.

Furthermore, in the filamentous fungus model Neurospora, numerous qiRNAs derived from the external and internal transcribed spacer regions (ETS, ITS1 and ITS2) have been described [66]. They are ~20–21 nt long with a strong occurrence of uridine (U) at their 5′ end and originate from both sense and antisense strands of the ribosomal DNA locus (Figure 2) [66]. In some cases, the biogenesis of qiRNAs requires the formation of double-stranded RNAs (dsRNAs), which is reminiscent of the structure of miRNA duplexes [53,66]. During their biogenesis, qiRNAs associate with the QDE-2 protein [53,66]. This and the 5′ and 3′ end nucleotide preferences for the rRFs generation suggest that qiRNAs are specific rRNA-derived RNAs (Figure 1) [66]. qiRNA expression requires DNA damage-induced aberrant RNAs (aRNAs) as a precursor, a process that depends on QDE-1 and QDE-3 function [66]. One potential role for qiRNAs in DNA damage response would be to inhibit protein translation [66].

According to a study conducted by Son et al. [82], it is now clear that mmu-miR-712 is generated upon preribosomal RNA cleavage by the exoribonuclease XRN1, which is involved in pre-rRNA maturation [77,82]. In mice, the mmu-pre-miR-712 sequence is embedded in the ITS2 region of the pre-rRNA [82]. The authors identified mmu-miR-712 as a negative regulator of tissue inhibitor of metalloproteinase 3 (TIMP3) expression. Furthermore, neutralizing mmu-miR-712 by anti-miR-712 rescues TIMP3 expression and prevents disease progression in murine models of atherosclerosis. Similarly to miR-712 in mice, a human-specific has-miR-663 could be derived from the spacer region of human RN45S gene (Figure 1 and Figure 2) [82]. 

Like mature rRNA, the ITS1 and ITS2 sequence length increased during evolution [130]. These lengthened sequences not only serve to recruit proteins and enzymes involved in rRNA biogenesis, but they may also harbor the sequence of functional ncRNAs, such as miRNAs or qiRNAs [66,82], and participate to post-transcriptional regulation or DNA damage response. This is why ITS1 and ITS2 should be studied, not only for their role in rRNA biogenesis, but also as template sequences for the biogenesis of small ncRNAs.

## 3. Conclusions

Taken together, the studies discussed in this review article demonstrate that the 28S, 18S, and 5.8S rRNAs, and even the ITS1 and ITS2, produce one or more small rRFs. These rRFs are present at various, yet significant, levels in different cell types or organs, and during development, like in the often described Zebrafish development model. The generation of these small rRFs does not appear to result from random degradation of the associated mature rRNAs. Moreover, the degradation rate of mature cytoplasmic rRNAs is generally beyond detection under normal conditions [131], as the rRNA is first fragmented by endoribonucleases and then the resulting by-products are rapidly degraded into mononucleotides by exoribonucleases [132,133]. Small rRF detection attests of their relative stability and implies that they do not result from normal cellular ribosome turnover. The caveat has to be taken into account that the study of rRFs has been hampered, and is still hampered, by the long-set bioinformatics pipelines that consider rRFs as mere degradation products and systematically remove small RNA sequencing reads mapping to rRNAs from the data [57,94]. 

Whereas the notion of degradation products is the first to come to mind when considering small rRFs, it was shown recently that tRNAs and rRNAs undergo stress-induced cleavage to produce stable rRFs products, and that this mechanism is conserved from yeast to human cells [107,111,113,114,116]. Interestingly, the best-described rRFs in the literature were found to be associated with proteins, such as Ago, PIWI, or PHAS proteins, suggesting that they function as part of RNP complexes. While their functional and biological significance remains to be fully appreciated, we might want to speculate that some rRFs may play a role in rRNA processing/degradation as well as in miRNA-like pathways. Therefore, we may want to consider rRNA both as a central player in the mRNA translational machinery, as a constituent of ribosomes, as well as a cleavable precursor of small rRFs, which may be considered as a novel class of potentially functional sRNAs that may regulate gene expression post-translationally.

It is likely that these rRFs are not generated in a random manner, but by a highly regulated cleavage and maturation process. Knowing that they derive from a highly expressed rRNA precursor, the rRF generation could be under strong selective pressure. It may be that the biogenesis of a given rRF is guided by RNA motifs and the recruitment of specific RNA binding proteins [65,67]. 

Moreover, some specific cleavages have been observed, especially for rRFs coming from the 3′ and 5′ ends of rRNAs [53]. In some cases, rRF3 or rRF5 have different expression patterns depending on the organisms [64], developmental stage or stress condition [27]. The rRFs originating from these central loci, however, are limited in number compared to the 3′ and 5′ ends. If rRF generation was neither directed nor regulated, they would be found more homogeneously distributed among RNA-sequencing data or not found at all due to degradation. 

Moreover, the same locus in two different organisms could generate two different rRFs involved in different functions [62]. That is already the case for rRFs coming from ITS1 or ITS2, which are sequences known to be highly heterogeneous [58,60,134]. For example, human ITS1 contains MER45C [135,136], while this region contains a miRNA in Drosophila. Moreover, in the 45S rDNA repeats, there is sequence variability [10]. Therefore, the variability of rRF sequences may also depend on which 45S rDNA precursor they derive from. Given the number of rDNA repeats in the genome, such mechanism is probable and would possibly lead to different effects in the same species. 

A comparable mechanism could be imagined for the rRFs deriving from mature 18S, 28S, and 5.8S rRNAs, but, so far, there is little known about the sequence variability of these rRNAs [62]. 

In both cases, these rRFs could derive from mature and functional rRNAs that would be differentially processed depending on the environmental condition. Another possibility is that they would be cleaved from certain rRNA pseudogenes that are not matured and incorporated into ribosomes, but instead serve as precursors for rRFs.

Finally, when looking at the targets of these rRFs, there is evidence that some of these act like miRNAs and target mRNAs to prevent their expression by inhibiting their translation or leading them to degradation [65,66,67,84,86,129]. Other evidence pinpoint rRFs as DNA-binding molecules that prevent the translocation of transposons [44]. In both cases, these molecules are possibly key contributors to the expression and stability of the genome. Therefore, there may not be a specific role for these rRFs, but more likely a variety of functions that depend on the species, cell type, and environmental conditions.

## Figures and Tables

**Figure 1 ncrna-05-00016-f001:**
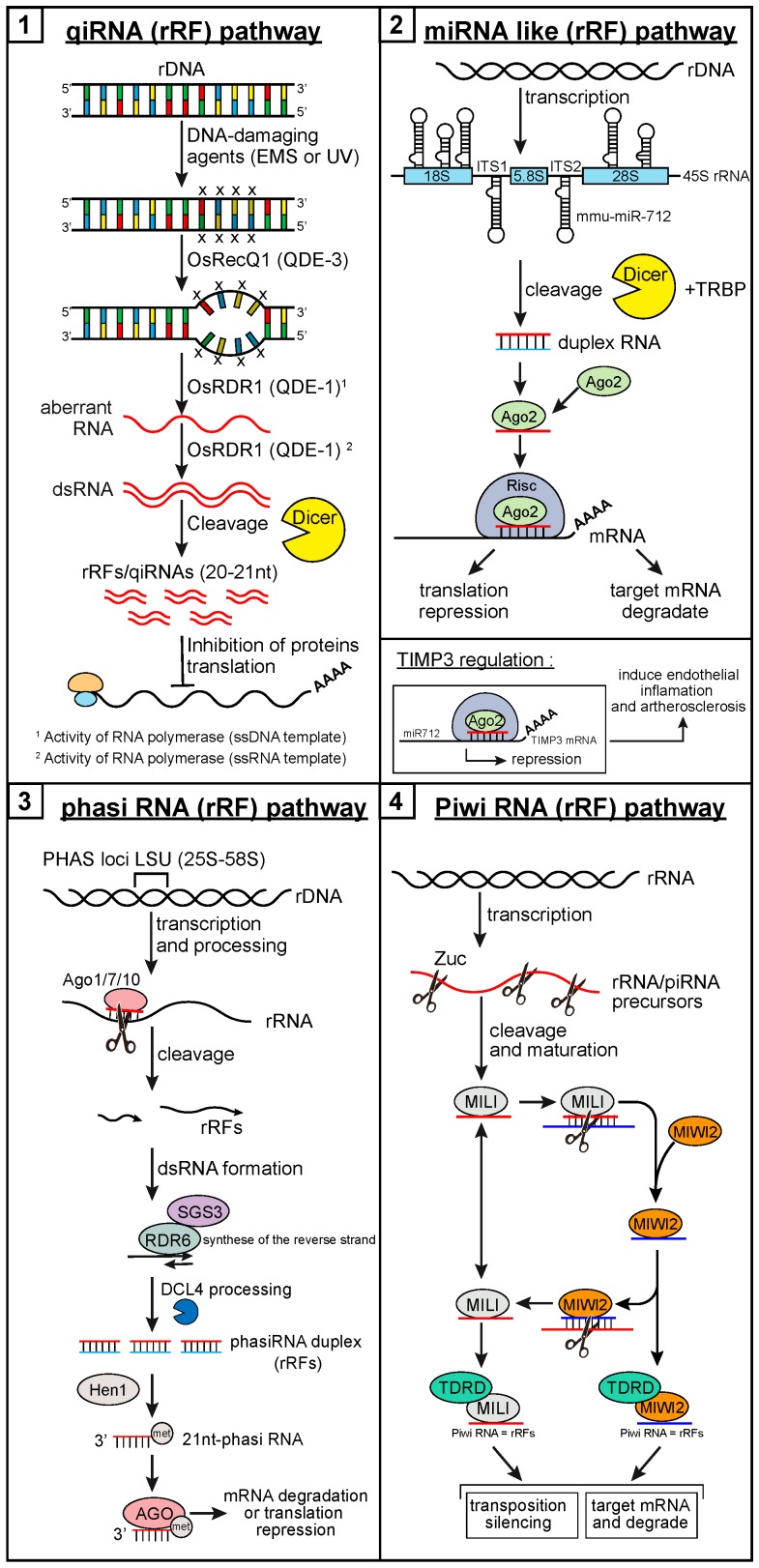
The biogenesis and function of ribosomal RNA-derived fragments. (**1**) QDE-2-interacting small RNAs (QiRNA)/ribosomal RNA-derived fragment (rRF) pathway discovered in fungi (*Neurospora crassa)* [66], and recently found in plants, flies, and mammals [72,79,80,81]. These rRFs originate from ribosomal DNA (rDNA) after DNA damage, which is detected by OsRecQ1 (RecQ DNA helicase homologue/QDE-3). This leads to recruitment of OsRDR1 (RNA-dependent RNA polymerase [RdRp] homologue/QDE-1) at the single-stranded DNA (ssDNA) site, production of aberrant RNA (aRNA) from ssDNA, and conversion of the aRNA into double-stranded RNA (dsRNA) via its RdRp activity. Dicer processes the dsRNA substrate into qiRNA rRFs, which then serves as guide RNA to repress messenger RNA (mRNA) translation. (**2**) Native ribosomal RNAs (rRNAs) harbor microRNA (miRNA) sequences, which may be generated under specific conditions (e.g., stress). These miRNAs may be located in internal transcribed spacer (ITS1), as hsa-miRNA-663 in humans [65], or in ITS2, as mmu-miRNA-712 in mice [82]. In Opium poppy, two and three miRNAs are present in the 18S and 28S rRNAs, respectively [83]. These miRNAs/rRFs follow the noncanonical miRNA pathway and repress translation of its mRNA targets. For example, in mice, tissue inhibitor of metalloproteinase 3 (TIMP3) mRNA is repressed by mmu-miR-712. TIMP3 being an inhibitor of MMP2/9 (matrix metalloproteinase-2/9) and of ADAM 10/12 (disintegrin and metalloproteinase 10/12) expression [82], its repression induces endothelial inflammation and atherosclerosis. (**3**) In the phased small interfering RNAs (phasiRNA)/rRF pathway, the large subunit (LSU) loci of rDNA are transcribed into phasiRNA precursors (pre-phasiRNAs). A miRNA incorporated into Ago1 (or 7 or 10) effector complexes guides endonucleolytic cleavage of the pre-phasiRNA [84], generating two rRFs, one of which acts as an RDR6 template, leading to the production of dsRNA. DCL4 processes the dsRNA, and produces phasiRNAs that are methylated (Met) by HEN1 [85]. Once incorporated into Ago1-loaded RNA-induced silencing complex (RISC), phasiRNAs/rRFs (21 nt) guide cleavage of homologous mRNAs [86], illustrating the importance and biological significance of rRFs. (**4**) In the P-element induced wimpy testis (PIWI)–piRNA/rRF pathway, some piRNA/rRF precursors are produced from rDNA. In the primary processing pathway, piwi-interacting RNA (piRNA) precursor are transcribed, exported to the cytoplasm, processed by Zuc and methylated by the methyltransferase Hen1 [87]. The resulting mature piRNAs are selected and loaded onto MILI (piwi-like protein 2, or PIWIL2) protein (in mouse, PIWI or AUB [aubergine] in Drosophila), which can enter the secondary processing pathway (the ping-pong cycle). MILI-piRNA/rRF complexes mediate cleavage of piRNA precursors and transposon (and protein-coding) transcripts, which silences transposon and gene expression at the post-transcriptional level [88]. These cleavage products are then loaded onto MIWI proteins (in mouse, Ago3 in Drosophila), which share functional features with MILI-piRNA/rRF complexes. The piRNA biogenesis pathways are well conserved across species, such as *C. elegans*, fish, and mouse. MILI/PIWI–piRNA complexes are involved in translational regulation by interacting with polysomes [89], mRNA cap-binding complex (CBC, in mice), and mRNA deadenylase (DeA, in Drosophila) [90]. MILI/PIWI proteins and piRNAs regulate the expression of genes and transposons at both transcriptional and post-transcriptional levels. EMS, ethyl methanesulfate; UV, ultraviolet.

**Figure 2 ncrna-05-00016-f002:**
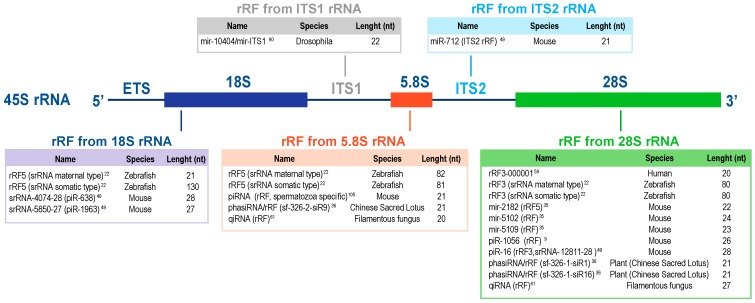
List of the major rRFs reported in the literature. The rRFs have been classified beyond the provenance from the 18S, 5.8S, 28S, ITS1, or ITS2 rRNA. For each rRF, the name, species of origin, and length are specified. An rRF described in a species may not be conserved in other species. Each organism has different levels of regulation and organization, and, although similarities exist between organisms, the occurrence and sequence of rRFs in a given rRNA region may be different. For example, ITS1 contains MER45C, which may be specific to vertebrates. Yet, in Drosophila a microRNA is shown in that region.

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
