# Peer review of "Small Non-Coding RNAs Derived from Eukaryotic Ribosomal RNA"

_ncrna, 2019, doi:10.3390/ncrna5010016_

Round 1

Reviewer 1 Report

This is an excellent review on eukaryotic rRNA derived small ncRNAs from Dr. Patrick Provost’s group which highlights several important and convincing aspects of the biogenesis and function of these unique ncRNAs. There are not many well organized and nicely written review like this one for rRF category of small RNAs, therefore this manuscript merits publication without a lot of changes. This reviewer has some minor comments that authors might want to take a look at before an editorial decision is made:

1. Abstract: Part of the abstract focuses on long standing controversy on the notion that small RNAs derived from rRNAs could be degradation products. However, now there are convincing molecular and biochemical evidences suggesting that these rRFs are real. Since these evidences are already in the body/main text of the manuscript, why not highlight the NUMBERS of rRFs that are experimentally verified and their association with known regulatory proteins in the abstract itself to make the case more convincing?

2. Line 117: “The rRF3 series were significantly more…” - this sounds like a random start on a series of rRFs. May be give a short introduction re how many rRFs are in this series and then talk about why rRF3 is significant.

3. Line 144 and line 155: There might be some syntax issues with how these words are used: ‘reverse-complement bind’ and ‘reverse-complementary bind’. Please check. Could it be: Reverse-complementarily bind ?

4. Lines 178-183: starting from  “An involvement of these rRFs…” cite References.

5.  Lines 315-316: “These helices may be recognized…” - is this a hypothesis or it is referencing a work already done? Not clear from the text.

6. Line 347: Title font size is different than rest of the text.

7. This reviewer appreciates the discussion provided in the ‘Conclusions’ section, which is primarily based on what is described in the main text. However, authors could also consider some outstanding thoughts and questions that might grab attention of curious readers. For example, are these rRFs under any selection pressure given that the number of  rRNAs and the rRFs originating from these loci are limited in number, compared to rRF targets (example: mRNAs) for gene regulation? Are there rRFs originating from rRNA pseudogenes?   

Author Response

Please refer to the uploaded PDF file.

Reviewer 2 Report

Minor requirements:

- cut the introduction into smaller chapters with headlines

- page 5 line 196: please explain qiRNA

- page 11 line 423: please explain rsRNA

- page 12 line 445/446: "Moreover, Dicer was found within the nucleolus and bound to both active and inactive copies of the rRNA gene (ref 129)". The ref 129 did not indicate that Dicer is indeed located within nucleoli.

Author Response

Please refer to the enclosed PDF document.

Reviewer 3 Report

In this review, the authors examine the literature of the last decade and assemble together pieces of evidence supporting a hypothesis that fragments of ribosomal RNAs may be meaningful entities and not degradation products. The manuscript is generally well-written and covers a lot of ground, as expected from a Review article. However, several things will need to be addressed before the manuscript can be accepted for publication.

• The current title is misleading and will need to be edited because neither the biogenesis nor the function of these fragments are known. Listing a few disconnected pieces of evidence does not amount to an understanding of how either works. While we know a lot about the biogenesis and function of the individual rRNAs, this is a review about short fragments from these RNAs and virtually nothing is known about how these fragments are made or what they do. My suggestion is to remove "Biogenesis and function of" from the title entirely.

• A prevalent feature of the narrative is a tendency to extrapolate from a few datapoints that appear in different papers and different organisms, turning them into a concrete statement. Here is an example: (line 388-90) "The biogenesis of qiRNAs requires the formation of double-stranded RNAs (dsRNAs) reminiscent of the structure of miRNA duplexes (49,61)." I do not believe that the authors of this earlier paper make such an unambiguous statement anywhere in their article. Such generalizations could lead a reader from a different field into believing that the statement is true beyond a doubt and that the matter has been resolved. It will be important that the authors simply state the evidence and avoid generalizing. These generalizations occur at multiple locations in the text. 

• I am not aware of a publication that has studied the distribution of lengths for fragments from rRNAs in any systematic manner. Consequently, any statements about length distributions of fragments from rRNAs should be qualified.

• I found several oversights in the presentation. Here is an example: (lines 359-60)  "Concerning expression patterns, rRNA-derived sequences ... than tRFs in human and mouse cells(100)." It seems to me that Ref. 100 is about 'wheat' and not about human and mouse tissues. Another example appears on line 313: "Interestingly, ... 28S rRNA(102)." Having read the paper of Ref. 102, I do not recall such a statement in that publication. If the statement is based on the authors' interpretation of the data presented in (102), this should be stated in this Review. Yet another example is the claimed lack of dependence on Dicer (see below).

• The section about the links to Argonaute and the RNAi pathway is underdeveloped. Assuming that this interaction indeed happens, the authors take the position that these rRNA fragments act like "miRNAs." Is it not possible that they might be the targets of a miRNA instead?  See for example Atwood et al JBC 2016, which also provides evidence of dependence on Dicer.  Notably, this paper by Atwood and other key papers e.g. Zhou Nat Struct Mol Biol 2017 are not discussed in this Review, and should be included.

• There are multiple instances where the narrative does not explicitly state the organism to which a statement refers. For example, line 395 mentions "miR-712."  A reader from a different field might think that miR-712 is a human miRNA whereas in reality it is a mouse miRNA.

• In several places, the authors mention that tRNA fragments are typically 20 nts long. This is not correct: see e.g. Telonis et al Oncotarget 2015; Pliatsika et al NAR 2017, etc. for different length distributions that change between tissues, and between health and disease. Related to this, the authors state that "tRNA cleavage into tRFs has often been described as a stress-dependent phenomenon."  This is incorrect in that it lumps together 'tRNA halves' that are produced upon stress, tRNA halves that are endogenously present even in unstressed cells, and the shorter tRNA fragments or tRFs that are also endogenously present.

• Figure 2 tries to generalize the findings but does not account for the fact that for example ITS1 contains MER45C which I believe is specific to vertebrates. Yet, a miRNA from Drosophila is shown in that region. The authors should emphasize in the text that although similarities exist between organisms there are also differences that may be relevant. In the absence of a systematic study and comparison of rRNA-derived fragments across organisms it is important to emphasize that differences may also exist.

• There are many places where the authors state "this suggests that the rRNA-derived RNAs are not degradation products." Given that this is a Review article, this is best mentioned in the abstract and the Discussion, and not in the main presentation.

Author Response

Please refer to the uploaded PDF file.

Reviewer 4 Report

The manuscript gives a pretty complete overview of the current knowledge on eukaryotic rRNA fragments (rRFs). It is well organized with a description of the knowledge on rRFs that originate from the different regions of the rRNA precursors or mature rRNA; it includes various subsections that describe the discovery, the sequence and structure and the function. I think the manuscript should be published. You will find below some comments you may want to address to improve some details of the manuscript. 

I completely subscribed to the statement in the abstract to encourage scientists to look at the "unexpected" and unexplored places (lines 20-23). In the introduction, the authors explain briefly one of the reasons when the rRNA are simply removed from RNA-seq analyses (lines 88-93). However, we may still miss some rRFs even including rDNA sequences (end of the introduction) because standard approaches use a reference transcriptome that does not include all the possible variations from rDNA or rRNA sequences. So, the authors may want to include a comment on the future challenges to detect rRFs that actually differ from the rDNA or rRNA sequences used as a reference transcriptome. Some methods actually exist to identify transcript variants without using any reference. So, it coudl be relevant to be a bit more precise when referring to variations and give a more quantitative indication of the observed variations. (line 42). 

Actually, rRNAs are just another kind of RNA subjected to RNA decay; some recent data suggest show the existence of differentially decaying fragments from mRNA transcripts in prokaryotes (Dar & Sorek, 2018). The more abundant fragments have a conserved structure which is a feature that may also explain the presence of specific rRFs (in particular in the 18S, 5.8S and 28S rRNAs). The authors may give more details on the rRFs that have a conserved 2D structure for example, or some conserved RNA binding sites. 

Author Response

Please refer to the uploaded PDF file.

Non-Coding RNA EISSN 2311-553X Published by MDPI AG, Basel, Switzerland RSS E-Mail Table of Contents Alert
Back to Top